# Role of oral health in heart and vascular health: A population-based study

**Amr Sayed Ghanem**[1], **Orsolya Németh**[2], **Marianna Móré**[3], **Attila Csaba Nagy**[1]*

**1** Department of Health Informatics, Institute of Health Sciences, Faculty of Health Sciences, University of Debrecen, Debrecen, Hungary, **2** Department of Community Dentistry, Faculty of Dentistry, Semmelweis University, Budapest, Hungary, **3** Institute of Social and Sociological Sciences, Faculty of Health Sciences, University of Debrecen, Nyíregyháza, Hungary

* nagy.attila@etk.unideb.hu

## Abstract

### Background and aim

Conditions such as hypertension, cardiovascular diseases, and hypercholesterolemia, are a major public health challenge. This study investigates the influence of oral health indicators, including gum bleeding, active dental caries, tooth mobility, and tooth loss, on their prevalence in Hungary, considering socioeconomic, demographic, and lifestyle factors.

### Materials and methods

Data from the 2019 Hungarian European Health Interview Survey with 5,603 participants informed this analysis. Data were accessed from the records maintained by the Department of Health Informatics at the University of Debrecen between September and November 2023. Variable selection employed elastic net regularization and k-fold cross-validation, leading to high-performing predictors for weighted multiple logistic regression models. Sensitivity analysis confirmed the findings' validity.

### Results

Significant links were found between poor oral health and chronic cardiac conditions. Multiple teeth extractions increased hypertension risk (OR = 1.67, 95% CI: [1.01–2.77]); dental prosthetics had an OR of 1.45 [1.20–1.75]. Gum bleeding was associated with higher cardiovascular disease (OR = 1.69 [1.30–2.21]) and hypercholesterolemia risks (OR = 1.40 [1.09–1.81]).

### Conclusions

Oral health improvement may reduce the risk of cardiac conditions. This underscores oral health's role in multidisciplinary disease management.

**Data Availability Statement:** The data presented in this study are not available to the public but can be requested from the institution that performed and supervised the data collection and primary

analysis: Hungary's Central Statistical Office, www.ksh.hu/?lang=en.

**Funding:** The author(s) received no specific funding for this work.

**Competing interests:** The authors have declared that no competing interests exist.

# 1. Introduction

## 1.1 Impact of heart and vascular conditions globally and in Hungary

Chronic non-communicable diseases (NCDs) remain a major challenge in global public health, responsible for 74% of all deaths in 2019. Among these, heart and vascular conditions such as hypertension and cardiovascular diseases (CVDs) are increasingly prevalent and significant. Ischemic heart disease, a key CVD, accounted for 16% of global fatalities. The rising incidence of heart and vascular conditions, also reflected in conditions like stroke and chronic obstructive pulmonary disease, underscores the critical need for targeted public health strategies [1–3].

Within this context, hypercholesterolaemia presents as a variable of particular significance, contributing to the plot of CVDs. Characterized by perturbations in lipid profiles—be it elevated total cholesterol, low-density lipoprotein cholesterol (LDL-c), or triglycerides (TG)—hypercholesterolaemia has long been implicated in cardiovascular pathology [4, 5].

Europe leads with a 53.7% prevalence of hypercholesterolemia, contrasted by much lower rates in regions like Southeast Asia and Africa [6].

Hypertension affects an estimated 1.28 billion adults worldwide, particularly in low- and middle-income countries. Interestingly, only 46% of these individuals are aware of their condition, and just 21% effectively manage their hypertension. The etiological landscape is further complicated by many of factors, including genetic predispositions and lifestyle choices such as obesity and excessive sodium intake. CVDs, including coronary and cerebrovascular diseases, are the leading global cause of death, accounting for 17.9 million fatalities in 2019. Once again, low- and middle-income countries bear the brunt of this condition. While traditional risk factors have been heavily studied, evidence posits a compelling link between oral health and the onset or exacerbation of CVDs [7–10].

In Hungary, NCDs present an alarming public health situation, with 40% of adults reporting at least one chronic condition, surpassing the EU average of 36%. The nation leads the EU in preventable mortality rates at 326 per 100,000 individuals, primarily due to lung cancer, ischaemic heart disease, and lifestyle risks like smoking. CVDs account for one-third of all deaths, and hypertension prevalence stands at 34% for women and 29% for men [11–13].

## 1.2 Oral health: The overlooked factor in chronic cardiac conditions

Globally, oral health conditions, affecting about 3.5 billion individuals, exceed the combined prevalence of the top five non-communicable diseases. Nearly 19% of adults, or over 1 billion people, suffer from severe periodontal diseases, while dental caries in permanent teeth impact 2 billion [14–16].

In Hungary, the oral health situation presents notable challenges linked to chronic diseases. Leading in dental caries prevalence in Europe, Hungary also grapples with significant rates of edentulism among the elderly; 30% of those aged 65 and above, and 40% of those over 75, have experienced complete tooth loss. This reflects not only poor oral health but also broader social and economic disparities. Additionally, gingivitis and periodontitis affect about a third of the population. These issues highlight the critical need to address Hungary's unique oral health concerns within the broader European context [17–22].

## 1.3 The link between oral health and heart-related conditions

Previous research has highlighted the link between periodontal health, CVDs, and hypercholesterolemia, with a focus on Porphyromonas gingivalis (P. gingivalis) [23, 24]. This pathogen is implicated in atherosclerosis development by oxidizing HDL and LDL, damaging vascular

endothelial cells, and contributing to endothelial dysfunction. P. gingivalis also promotes the proliferation of smooth muscle cells, formation of foam cells, and destabilization of atherosclerotic plaques, exacerbating cardiovascular outcomes [25]. A key mechanism involves the secretion of gingipains, cysteine proteases that degrade host proteins, incite inflammation, impair vascular function, and promote clot formation through platelet activation and fibrinogen degradation. These proteases also activate matrix metalloproteinases (MMPs), which contribute to plaque destabilization. Additionally, P. gingivalis invades endothelial cells, inducing oxidative stress and elevated reactive oxygen species (ROS) levels, further aggravating endothelial dysfunction and increasing cardiovascular risk [26–30].

As the preceding discussion predominantly emphasized the links between periodontitis and cardiovascular diseases, it is imperative to consider the early-stage oral health markers that constitute the focal point of this study. Dental caries, as identified by Folayan et al. (2021), not only serve as plaque retention factors but also foster biofilm proliferation, conducive to anaerobic bacteria growth and gingival inflammation [31]. Further, Sabharwal et al. (2021) linked dental caries to an increased risk of metabolic diseases [32]. These findings highlight the potential role of chronic oral inflammation and salivary biomarkers in mediating the link between dental caries and systemic metabolic disorders.

Gingival bleeding serves as both a primary symptom and an initial clinical indicator of periodontal disease [33]. It is frequently used by clinicians as a hallmark of active disease, highlighting its relevance in diagnosing the inflammatory state of the periodontium [34]. Additionally, the progression of periodontal disease can lead to tooth mobility, resulting from the degradation of the tooth's supporting structures like the periodontal ligament and alveolar bone. This process typically involves bone resorption and gingival recession, further destabilizing teeth [35].

Tooth loss and edentulism are recognized as markers for chronic conditions. Sara Hag Mohamed and Wael Sabbah's 2023 study identified tooth loss as an early indicator of chronic diseases in working-age adults, linked to shared risk factors affecting both oral and general health [36]. Similarly, Yuqing Zhang et al. (2022) found a strong association between edentulism, tooth loss, and multiple chronic diseases. Their research suggests that individuals with multiple chronic conditions are more likely to be edentulous than those with fewer or no chronic diseases, underscoring the significance of tooth loss and edentulism in evaluating the relationship between oral health and systemic health [37].

In Central Eastern Europe, especially in Hungary, there is a notable lack of research that simultaneously considers a large, representative sample and a broad set of oral health variables, such as gum bleeding, dental caries, tooth mobility, and tooth loss, to investigate the relationship between oral health and heart related conditions. While Hungary faces high rates of cardiovascular diseases and their related conditions, and some of Europe's worst oral health indicators, previous studies have not fully explored how social, demographic, economic, and lifestyle factors might confound this relationship. This study, utilizing data from the 2019 Hungarian instalment of the European Health Interview Survey (EHIS), aims to fill this gap. It explores the connections between oral health and heart and vascular health, considering the influence of a range of confounding variables.

## 2. Materials and methods

### 2.1 Study design

The EHIS is a representative survey conducted every five years in each member state of the EU, and covers a range of health variables, extending from lifestyle habits to chronic diseases, with a specific segment on oral health metrics [38].

To account for non-response bias and maintain a representative sample, individualized weighting was applied. This method followed Eurostat recommendations to synchronize the sample's demographic profile with the general population [39].

Managed by the Hungarian Central Statistical Office under Eurostat's supervision, the dataset was obtained using a rigorous stratified sampling technique, ensuring its generalizability to Hungary's adult populace residing in households.

This research is confined to data from the 2019 EHIS cycle, and was performed in line with the principles of the Declaration of Helsinki. Approval was granted by the Ethics Committee of the University of Debrecen (5609–2020). All data used in this study were fully anonymized before access.

## 2.2 Sample population and data collection

The survey targeted a representative sample from across Hungarian municipalities, achieving a final response rate of approximately 47% from the intended 12,002 participants. Additional data were collected for children residing in sampled households. Data acquisition, detailed in [40] combined electronic and in-person methods using a standardized Eurostat questionnaire, ensuring consistency and cross-national comparability. The data collection period spanned from September 16 to December 31, 2019. The dataset for this study was accessed from the records maintained by the Department of Health Informatics at the Faculty of Health Sciences, University of Debrecen. Data retrieval was conducted over a period from September to November 2023.

## 2.3 Data treatment and variable specification

The study utilized key oral health indicators, including self-perceived oral health (categorized as 'Average,' 'Good,' or 'Bad'), number of unreplaced teeth extracted due to decay ('None,' '1 to 5,' '6 to 19,' 'More than 20'), filled teeth, active dental caries, tooth mobility, and gingival bleeding. Overall oral health was assessed as 'Optimal' or 'Suboptimal,' and the timing of the last dental visit was noted. Sociodemographic variables (age, gender, residence area, employment status, and education level) and lifestyle factors (alcohol consumption, smoking, BMI) were also included. Participants' financial standing was evaluated both subjectively and within quintiles. Composite variables were created for CVDs (encompassing stroke, myocardial infarction, arrhythmia, and coronary artery diseases) and cardiovascular risk profile (combining hypertension, CVDs, and hypercholesterolemia) to provide a comprehensive assessment of cardiac health. The main outcomes investigated were binary classifications of these conditions [41].

## 2.4 Variable selection method and statistical analysis

A two-stage variable selection strategy was employed to optimize model reliability and accuracy. The initial stage involved applying weighted Pearson's Chi-square tests to potential predictor variables to assess their association with each chronic disease outcome, alongside obtaining weighted proportions. Variables were chosen based on their documented impact on the outcomes studied within the scientific literature, ensuring a solid theoretical foundation. This selection was further supported by empirical evidence, underscoring the relevance and significance of each variable included in the analysis. Subsequently, a penalized regression approach was employed using Elastic Net regularization, a hybrid of Lasso and Ridge regression techniques. Elastic Net incorporates a hyperparameter $\alpha$ set at 0.5, allowing for a balanced compromise between Lasso's variable selection properties and Ridge's ability to handle multicollinearity. This approach effectively mitigates the risks of overfitting and multicollinearity while facilitating more accurate variable selection [42–44].

K-fold cross-validation was conducted to iteratively partition the dataset and gauge the predictive accuracy of the penalized model across multiple data folds. Variables that consistently exhibited high predictive performance were selected for inclusion in the final model. It is noteworthy that the hybrid Lasso-Ridge approach was compared against a stepwise logistic regression model and was found to surpass it in terms of model fit and strength, as evidenced by lower values in information criteria such as the Akaike Information Criterion (AIC) and the Bayesian Information Criterion (BIC).

Upon finalization of the predictor set, weighted multiple logistic regression models were constructed. Their performance was subsequently validated through a series of diagnostic tests, confirming both their goodness-of-fit using the Hosmer–Lemeshow test, and predictive strength.

The level of statistical significance was set at a p-value below 0.05. This criterion was applied across all analytical procedures to ascertain the meaningfulness of relationships between variables. Results stemming from logistic regression were expressed in terms of odds ratios (ORs), accompanied by 95% confidence intervals (CIs). The entirety of the statistical evaluations was executed using STATA IC Version 17.0 software [45].

## 2.5 Sensitivity analysis

Robustness tests were conducted through bootstrapping with 1,000 repetitions and comprehensive subpopulation analyses. The bootstrapping process involved applying the logistic regression model across various demographic, lifestyle, and oral health variables to assess the stability of the findings under varied sampling conditions. Concurrently, subpopulation analyses were performed across different demographic groups defined by age, gender, educational status, employment status, area of residence, and income levels. These analyses aimed to evaluate the consistency of the associations identified in the primary analysis within diverse segments of the study population.

# 3. Results

## 3.1 Data completeness

The study analyzed a total sample of 5,603 participants. The completeness of data for each variable of interest varied as follows:

- Variables with complete data (5,603 observations): Age groups, Gender, Area of residence, Educational attainment, Employment, Smoking status, Alcohol use, Oral health, Income quintiles, Cardiovascular risk profile.

- Variables with high data availability: Self-perceived health (5,570), Hypertension (5,556), Last dental checkup (5,520), BMI (5,537), and Presence of prosthetic tooth replacement (5,534).

- Variables with moderate data availability: Financial status (5,467), Hypercholesterolemia (5,462), CVDs (5,594), Presence of chronic disease (5,489), Has filled teeth (5,486), Has permanent teeth missing due to decay (extracted) and not replaced (5,476), Has mobile Teeth (5,457), Gum bleeding (5,443), and Self-perceived oral health (5,562).

- Variables with lower data Availability: Has active caries (5,320) and Number of permanent teeth missing due to extraction (4,444).

This overview indicates a high level of data completeness across most variables. Notable exceptions include presence of active caries and missing teeth except wisdom, where a

relatively larger proportion of data was missing. The potential impact of these missing data on the study's findings is discussed further in the 'Limitations' section.

### 3.2 Sociodemographic and lifestyle characteristics

As seen in Table 1, gender differences were significant, with women exhibiting higher prevalence rates for hypertension (17.71%) and CVDs (8.7%), and a notable prevalence of hypercholesterolemia (7.66%; all p<0.05). Among age groups, the 65+ demographic showed increased prevalence in hypertension (15.32%) and CVDs (7.87%; p<0.001), as well as a noticeable rate in hypercholesterolemia (6.02%; p<0.001).

Socioeconomic factors, such as educational attainment and employment status, also had a marked impact. Lower educational attainment was linked to higher rates of hypertension (16.54%) and CVDs (7.93%), while unemployment correlated with higher prevalence in all three conditions (hypertension: 19.85%, CVDs: 10.42%, hypercholesterolemia: 7.77%; p<0.001). Lifestyle factors, specifically non-smoking status and overweight/obese condition consistently showed higher prevalence rates across all outcomes (hypertension: 24.35% and 24.83%, CVDs: 10.27%, hypercholesterolemia: 10.16% and 10.53%; p<0.001). Alcohol consumption showed marginal significance in hypercholesterolemia (8.94%; p = 0.049).

### 3.3 Comparative oral health profiles

Results in Table 2 showed significant links between hypertension, self-perceived oral health, dental checkup frequency, and presence of filled teeth (all p < 0.001). Hypertensive individuals reported worse self-perceived oral health and had fewer dental checkups compared to their non-hypertensive counterparts. In the context of CVDs, individuals with CVDs were more likely to report poor oral health, have fewer dental checkups, have a higher number of teeth missing due to extraction, and more likely to have prosthetic replacements (all p < 0.001).

Hypercholesterolemia was significantly associated with suboptimal self-perceived oral health, gum bleeding, infrequent dental checkups, fewer filled teeth, and higher mobility and number of missing teeth (p values ranging from <0.001 to 0.038).

### 3.4 Multiple logistic regression models

Tables 3 and 4 present multiple logistic regression analysis for the key outcomes of interest. For hypertension, younger age groups (15–34: OR = 0.14 [0.09–0.21]; 35–64: 0.45 [0.35–0.57]) and higher educational attainment (secondary: 0.73 [0.60–0.90]; tertiary: 0.63 [0.48–0.83]) were associated with lower odds. Normal BMI (0.42 [0.35–0.51]) and non-drinking (0.80 [0.66–0.97]) also reduced odds, whereas oral health factors like 6 to 19 extracted teeth (1.67 [1.00–2.77]) and prosthetic replacements (1.45 [1.20–1.75]) increased them.

For CVDs, younger ages (15–34: 0.40 [0.23–0.70]; 35–64: 0.64 [0.49–0.84]) and employment (0.69 [0.52–0.91]) were protective, while chronic disease presence (4.34 [3.18–5.92]), poor self-perceived health (2.70 [2.12–3.45]), gum bleeding (1.69 [1.30–2.21]), and prosthetic teeth (1.35 [1.07–1.70]) increased risks.

In hypercholesterolemia, younger age (15–34: 0.27 [0.15–0.50]; 35–64: 0.63 [0.48–0.83]), urban residence (1.28 [1.03–1.59]), and tertiary education (1.40 [1.03–1.89]) showed significant associations. Normal BMI (0.51 [0.40–0.65]) was protective, while chronic disease (3.16 [2.39–4.17]), poor self-perceived health (1.67 [1.29–2.16]), gum bleeding (1.40 [1.09–1.81]), prosthetic teeth (1.51 [1.19–1.92]), and recent dental checkups (1.51 [1.19–1.93]) correlated with increased risks. Key associations with cardiovascular risk profile outcomes were also identified. Younger age groups (15–34: 0.12 [0.09–0.17], 35–64: 0.41 [0.33–0.52]) had significantly lower odds than those 65+. Normal BMI (0.47 [0.40–0.56]) reduced odds and the presence of a

**Table 1. Sociodemographic and lifestyle characteristics of study participants, stratified by hypertension, cardiovascular diseases and hypercholesterolaemia.** Unweighted n (weighted %).

| Variable | Category | Hypertension | | | | Cardiovascular diseases | | | | Hypercholesterolaemia | | | |
|---|---|---|---|---|---|---|---|---|---|---|---|---|---|
| | | Without hypertension (n, %) | With hypertension (n, %) | Total (n, %) | P value | Without heart disease (n, %) | With heart disease (n, %) | Total (n, %) | P value | Without hypercholesterolaemia (n, %) | With hypercholesterolaemia (n, %) | Total (n, %) | P value |
| **Gender** | Male | 1702 (33.32) | 843 (13.79) | 2545 (47.11) | **<0.001** | 2187 (41.44) | 380 (5.77) | 2567 (47.21) | **<0.001** | 2171 (41.4) | 321 (5.52) | 2492 (46.92) | **0.004** |
| | Female | 1884 (35.18) | 1127 (17.71) | 3011 (52.89) | | 2478 (44.09) | 549 (8.7) | 3027 (52.79) | | 2510 (45.42) | 460 (7.66) | 2970 (53.08) | |
| **Age groups** | 65+ | 542 (7.84) | 1077 (15.32) | 1619 (23.15) | **<0.001** | 1070 (15.23) | 557 (7.87) | 1627 (23.1) | **<0.001** | 1188 (17.17) | 406 (6.02) | 1594 (23.19) | **<0.001** |
| | 15–34 | 1214 (25.97) | 49 (1.06) | 1263 (27.02) | | 1232 (26.23) | 42 (0.85) | 1274 (27.08) | | 1229 (26.72) | 26 (0.62) | 1255 (27.34) | |
| | 35–64 | 1830 (34.7) | 844 (15.12) | 2674 (49.82) | | 2363 (44.07) | 330 (5.76) | 2693 (49.82) | | 2264 (42.93) | 349 (6.54) | 2613 (49.47) | |
| **Area of residence** | Rural | 1166 (20.03) | 634 (9.71) | 1800 (29.74) | 0.232 | 1511 (25.1) | 299 (4.59) | 1810 (29.69) | 0.17 | 1571 (26.64) | 200 (3.14) | 1771 (29.78) | **<0.001** |
| | Urban | 2420 (48.47) | 1336 (21.79) | 3756 (70.26) | | 3154 (60.42) | 630 (9.89) | 3784 (70.31) | | 3110 (60.18) | 581 (10.04) | 3691 (70.22) | |
| **Educational attainment** | Primary | 1436 (25.71) | 1059 (16.54) | 2495 (42.25) | **<0.001** | 1975 (34.18) | 528 (7.93) | 2503 (42.11) | **<0.001** | 2119 (36.87) | 341 (5.53) | 2460 (42.4) | **<0.001** |
| | Secondary | 1247 (24.2) | 577 (9.48) | 1824 (33.68) | | 1601 (29.91) | 241 (3.84) | 1842 (33.75) | | 1553 (29.61) | 245 (4.17) | 1798 (33.77) | |
| | Tertiary | 903 (18.59) | 334 (5.48) | 1237 (24.07) | | 1089 (21.44) | 160 (2.71) | 1249 (24.14) | | 1009 (20.34) | 195 (3.49) | 1204 (23.83) | |
| **Employment** | Unemployed | 1445 (25.4) | 1336 (19.85) | 2781 (45.25) | **<0.001** | 2085 (34.72) | 709 (10.42) | 2794 (45.14) | **<0.001** | 2239 (37.66) | 504 (7.77) | 2743 (45.43) | **<0.001** |
| | Employed | 2141 (43.1) | 634 (11.65) | 2775 (54.75) | | 2580 (50.81) | 220 (4.05) | 2800 (54.86) | | 2442 (49.15) | 277 (5.42) | 2719 (54.57) | |
| **Financial status** | Average | 1890 (36.14) | 1225 (19.79) | 3115 (55.93) | **<0.001** | 2582 (47.24) | 550 (8.66) | 3132 (55.9) | **<0.001** | 2596 (48.02) | 464 (7.87) | 3060 (55.89) | **<0.001** |
| | Good | 1230 (25.19) | 435 (7.41) | 1665 (32.6) | | 1476 (29.3) | 194 (3.2) | 1670 (32.5) | | 1466 (29.47) | 183 (3.39) | 1649 (32.86) | |
| | Bad | 381 (.7.07) | 275 (4.41) | 656 (11.47) | | 497 (8.97) | 168 (2.63) | 665 (11.6) | | 522 (9.38) | 113 (1.87) | 635 (11.25) | |
| **Income quintiles** | First | 699 (12.81) | 449 (7.29) | 1148 (2.01) | **<0.001** | 928 (16.56) | 225 (3.49) | 1153 (20.05) | **<0.001** | 990 (17.86) | 142 (2.29) | 1132 (20.14) | **0.013** |
| | Second | 673 (12.43) | 496 (7.67) | 1169 (20.1) | | 937 (16.46) | 236 (3.57) | 1173 (20.03) | | 953 (16.87) | 190 (3.14) | 1143 (20.01) | |
| | Third | 710 (13.4) | 414 (6.5) | 1124 (19.9) | | 942 (16.91) | 196 (3.11) | 1138 (20.02) | | 960 (17.42) | 152 (2.59) | 1112 (20.01) | |
| | Fourth | 837 (15.72) | 420 (6.75) | 1257 (22.46) | | 1069 (19.51) | 197 (2.97) | 1266 (22.48) | | 1040 (19.28) | 197 (3.2) | 1237 (22.48) | |
| | Fifth | 667 (14.15) | 191 (3.29) | 858 (17.44) | | 789 (16.09) | 75 (1.33) | 864 (17.42) | | 738 (15.39) | 100 (1.96) | 838 (17.35) | |

*(Continued)*

**Table 1.** (Continued)

| Variable | Category | Hypertension | | | | Cardiovascular diseases | | | | Hypercholesterolaemia | | | |
|---|---|---|---|---|---|---|---|---|---|---|---|---|---|
| | | Without hypertension (*n*, %) | With hypertension (*n*, %) | Total (*n*, %) | P value | Without heart disease (*n*, %) | With heart disease (*n*, %) | Total (*n*, %) | P value | Without hypercholesterolaemia (*n*, %) | With hypercholesterolaemia (*n*, %) | Total (*n*, %) | P value |
| **Smoking** | Smoker | 1098 (20.85) | 426 (7.14) | 1524 (28) | **<0.001** | 1324 (24.58) | 213 (3.45) | 1537 (28.02) | **0.004** | 1350 (25.37) | 148 (2.65) | 1498 (28.02) | **<0.001** |
| | Non-smoker | 2488 (47.65) | 1544 (24.35) | 4032 (72) | | 3341 (60.95) | 716 (11.03) | 4057 (71.98) | | 3331 (61.45) | 633 (10.53) | 3964 (71.98) | |
| **BMI** | Overweight and Obese | 1781 (33.4) | 1532 (24.83) | 3313 (58.23) | **<0.001** | 2676 (47.93) | 655 (10.27) | 3331 (58.2) | **<0.001** | 2642 (47.7) | 598 (10.16) | 3240 (57.86) | **<0.001** |
| | Normal | 1770 (35.11) | 420 (6.66) | 2190 (41.77) | | 1938 (37.59) | 266 (4.22) | 2204 (41.8) | | 1996 (39.12) | 175 (3.02) | 2171 (42.14) | |
| **Alcohol use** | Drinker | 2590 (50.52) | 1273 (20.54) | 3863 (71.06) | **<0.001** | 3352 (62.6) | 545 (8.61) | 3897 (71.21) | **<0.001** | 3265 (62.01) | 527 (8.94) | 3792 (70.95) | **0.049** |
| | Non-drinker | 996 (17.98) | 697 (10.96) | 1693 (28.94) | | 1313 (22.92) | 384 (5.86) | 1697 (28.79) | | 1416 (24.81) | 254 (4.25) | 1670 (29.05) | |

Bold values indicate statistical significance, p<0.05 based on weighted Pearson's chi-squared test.

Table 2. Comparative oral health profiles of study participants with hypertension, cardiovascular disease and hypercholesterolaemia versus those without. Unweighted n (weighted %).

| Variable | Category | Hypertension | | | | Cardiovascular diseases | | | | Hypercholesterolaemia | | | |
|---|---|---|---|---|---|---|---|---|---|---|---|---|---|
| | | Without hypertension (n, %) | With hypertension (n, %) | Total (n, %) | P value | Without heart disease (n, %) | With heart disease (n, %) | Total (n, %) | P value | Without hypercholesterolaemia (n, %) | With hypercholesterolaemia (n, %) | Total (n, %) | P value |
| **Presence of active caries** | No | 2402 (48.16) | 1386 (22.94) | 3788 (71.1) | 0.065 | 3172 (60.71) | 631 (10.33) | 3803 (71.04) | 0.82 | 3191 (61.59) | 544 (9.62) | 3735 (71.21) | 0.1654 |
| | Yes | 1019 (20.35) | 488 (8.55) | 1507 (28.9) | | 1264 (24.82) | 253 (4.14) | 1517 (28.96) | | 1282 (25.33) | 193 (3.47) | 1475 (28.79) | |
| **Self-perceived oral health** | Average | 1023 (18.92) | 791 (12.43) | 1814 (31.35) | **<0.001** | 1489 (26.05) | 341 (5.35) | 1830 (31.4) | **<0.001** | 1475 (26.29) | 299 (4.84) | 1774 (31.13) | **<0.001** |
| | Good | 1976 (39.77) | 607 (10.3) | 2583 (50.07) | | 2346 (45.87) | 249 (4.13) | 2595 (50) | | 2301 (45.74) | 256 (4.71) | 2557 (50.44) | |
| | Bad | 567 (9.8) | 562 (8.78) | 1129 (18.58) | | 802 (13.62) | 334 (4.99) | 1136 (18.6) | | 881 (14.8) | 221 (3.63) | 1102 (18.43) | |
| **Has gum bleeding when brushing teeth** | No | 2925 (56.99) | 1622 (26.56) | 4547 (83.55) | 0.196 | 3852 (72.23) | 718 (11.3) | 4570 (83.53) | **<0.001** | 3870 (73.19) | 610 (10.52) | 4480 (83.71) | **0.012** |
| | Yes | 577 (11.59) | 291 (4.86) | 868 (16.45) | | 688 (13.32) | 185 (3.14) | 873 (16.47) | | 699 (13.72) | 147 (2.57) | 846 (16.29) | |
| **Last dental checkup** | More than a year ago | 1835 (34.01) | 1253 (19.8) | 3088 (53.8) | **<0.001** | 2486 (44.23) | 621 (9.6) | 3107 (53.83) | **<0.001** | 2601 (46.66) | 436 (7.17) | 3037 (53.83) | **0.007** |
| | Less than 6 months ago | 963 (19.71) | 395 (6.74) | 1358 (26.45) | | 1192 (23.57) | 175 (2.91) | 1367 (26.48) | | 1126 (22.56) | 212 (3.92) | 1338 (26.48) | |
| | less than a year but more than 6 months ago | 750 (14.88) | 292 (4.87) | 1042 (19.75) | | 926 (17.76) | 119 (1.93) | 1045 (19.69) | | 903 (17.65) | 120 (2.04) | 1023 (19.69) | |
| **Has filled teeth** | No | 1057 (19.47) | 838 (13.13) | 1895 (32.61) | **<0.001** | 1458 (25.71) | 439 (6.73) | 1897 (32.44) | **<0.001** | 1585 (28.09) | 288 (4.72) | 1873 (32.81) | **0.038** |
| | Yes | 2472 (49.15) | 1087 (18.24) | 3559 (67.39) | | 3122 (59.9) | 467 (7.66) | 3589 (67.56) | | 3020 (58.91) | 470 (8.29) | 3490 (67.19) | |
| **Has mobile teeth** | No | 3283 (64.59) | 1709 (27.91) | 4992 (92.5) | **<0.001** | 4231 (79.88) | 787 (12.6) | 5018 (92.48) | **<0.001** | 4255 (81.2) | 662 (11.45) | 4917 (92.66) | **<0.001** |
| | Yes | 225 (3.96) | 211 (3.54) | 436 (7.5) | | 321 (5.65) | 118 (1.87) | 439 (7.52) | | 325 (5.69) | 96 (1.66) | 421 (7.34) | |
| **Number of permanent teeth missing due to extraction** | None | 154 (4.08) | 40 (0.99) | 194 (5.08) | **<0.001** | 174 (4.6) | 20 (0.46) | 194 (5.05) | **<0.001** | 171 (4.59) | 17 (0.45) | 188 (5.03) | **<0.001** |
| | 1 to 5 | 1514 (38.42) | 533 (11.92) | 2047 (50.34) | | 1854 (45.86) | 206 (4.55) | 2060 (50.4) | | 1768 (44.67) | 245 (5.68) | 2013 (50.36) | |
| | 6 to 19 | 556 (12.38) | 666 (13.61) | 1222 (25.99) | | 918 (19.89) | 310 (6.14) | 1228 (26.03) | | 960 (20.82) | 246 (5.27) | 1206 (26.1) | |
| | More than 20 | 376 (7.28) | 586 (11.31) | 962 (18.59) | | 636 (12.3) | 326 (6.21) | 962 (18.51) | | 733 (14.33) | 208 (4.18) | 941 (18.52) | |

(Continued)

**Table 2.** (Continued)

| Variable | Category | Hypertension | | | | Cardiovascular diseases | | | | Hypercholesterolaemia | | | |
|---|---|---|---|---|---|---|---|---|---|---|---|---|---|
| | | Without hypertension (*n*, %) | With hypertension (*n*, %) | Total (*n*, %) | P value | Without heart disease (*n*, %) | With heart disease (*n*, %) | Total (*n*, %) | P value | Without hypercholesterolaemia (*n*, %) | With hypercholesterolaemia (*n*, %) | Total (*n*, %) | P value |
| Has permanent teeth missing due to decay (extracted) and not replaced | No | 1636 (33) | 685 (11.2) | 2321 (44.21) | <**0.001** | 2032 (39.24) | 302 (4.97) | 2334 (44.21) | <**0.001** | 2024 (39.65) | 267 (4.76) | 2291 (44.41) | <**0.001** |
| | Yes | 1891 (35.62) | 1236 (20.17) | 3127 (55.79) | | 2544 (46.41) | 601 (9.38) | 3145 (55.79) | | 2577 (47.29) | 491 (8.3) | 3068 (55.59) | |
| Presence of prosthetic replacement (bridge, denture, or implant) | No | 2232 (45.09) | 610 (10.33) | 2842 (55.42) | <**0.001** | 2563 (50.63) | 298 (4.84) | 2861 (55.47) | <**0.001** | 2574 (51.39) | 223 (4.12) | 2797 (55.52) | <**0.001** |
| | Yes | 1319 (23.41) | 1339 (21.17) | 2658 (44.58) | | 2054 (34.93) | 619 (9.6) | 2673 (44.53) | | 2060 (35.39) | 552 (9.1) | 2612 (44.48) | |
| Oral health | Optimal | 800 (16.64) | 171 (2.91) | 971 (19.55) | <**0.001** | 915 (18.55) | 68 (1.09) | 983 (19.64) | <**0.001** | 884 (18.28) | 78 (1.44) | 962 (19.73) | <**0.001** |
| | Suboptimal | 2786 (51.87) | 1799 (28.59) | 4585 (80.45) | | 3750 (66.97) | 861 (13.39) | 4611 (80.36) | | 3797 (68.53) | 703 (11.74) | 4500 (80.27) | |

Bold values indicate statistical significance, p<0.05 based on weighted Pearson's chi-squared test.

**Table 3. Weighted multiple logistic regression analysis of factors affecting hypertension and cardiovascular diseases.**

| Characteristics | | Hypertension | | Cardiovascular diseases | |
|---|---|---|---|---|---|
| | | OR (95% CI) | P value | OR (95% CI) | P value |
| **Gender** | Male | | | | |
| | Female | 1.13 [0.94–1.35] | 0.192 | 1.02 [0.83–1.26] | 0.828 |
| **Age groups** | 65+ | | | | |
| | 15–34 | **0.14 [0.09–0.21]** | **<0.001** | **0.40 [0.23–0.70]** | **0.001** |
| | 35–64 | **0.45 [0.35–0.57]** | **<0.001** | **0.64 [0.49–0.84]** | **0.001** |
| **Area of residence** | Rural | | | | |
| | Urban | | | | |
| **Educational attainment** | Primary | | | | |
| | Secondary | **0.73 [0.60–0.90]** | **0.003** | 0.91 [0.72–1.16] | 0.459 |
| | Tertiary | **0.63 [0.48–0.83]** | **0.001** | 1.28 [0.95–1.73] | 0.105 |
| **Employment** | Unemployed | | | | |
| | Employed | 0.89 [0.71–1.12] | 0.318 | **0.69 [0.52–0.91]** | **0.009** |
| **Financial status** | Average | | | | |
| | Good | 0.94 [0.76–1.15] | 0.524 | 1.16 [0.91–1.48] | 0.231 |
| | Bad | 0.93 [0.71–1.21] | 0.593 | 1.10 [0.84–1.45] | 0.491 |
| **Income quintiles** | First | | | | |
| | Second | 0.81 [0.63–1.04] | 0.101 | | |
| | Third | **0.70 [0.54–0.91]** | **0.008** | | |
| | Fourth | 0.93 [0.70–1.23] | 0.616 | | |
| | Fifth | 0.74 [0.53–1.03] | 0.073 | | |
| **Smoking** | Smoker | | | | |
| | Non-smoker | 1.08 [0.89–1.32] | 0.434 | | |
| **BMI** | Overweight and Obese | | | | |
| | Normal | **0.42 [0.35–0.51]** | **<0.001** | | |
| **Alcohol use** | Drinker | | | | |
| | Non-drinker | **0.80 [0.66–0.97]** | **0.02** | 1.04 [0.84–1.28] | 0.749 |
| **Has Chronic disease** | No | | | | |
| | Yes | **5.85 [4.80–7.13]** | **<0.001** | **4.34 [3.18–5.92]** | **<0.001** |
| **Self-perceived health** | Average | | | | |
| | Good | **0.74 [0.60–0.90]** | **0.003** | **0.45 [0.35–0.59]** | **<0.001** |
| | Bad | 1.11 [0.85–1.43] | 0.446 | **2.70 [2.12–3.45]** | **<0.001** |
| **Self-perceived oral health** | Average | | | | |
| | Good | 0.86 [0.71–1.05] | 0.141 | 1.01 [0.79–1.29] | 0.947 |
| | Bad | 0.86 [0.69–1.08] | 0.198 | 1.12 [0.88–1.42] | 0.367 |
| **Number of permanent teeth extracted due to decay, not replaced** | None | | | | |
| | 1 to 5 | 1.09 [0.67–1.77] | 0.74 | 1.12 [0.59–2.12] | 0.732 |
| | 6 to 19 | **1.67 [1.01–2.77]** | **0.049** | 1.49 [0.79–2.84] | 0.221 |
| | More than 20 | 1.35 [0.80–2.29] | 0.264 | 1.31 [0.69–2.51] | 0.407 |
| **Has filled teeth** | No | | | | |
| | Yes | | | 0.81 [0.64–1.03] | 0.088 |
| **Presence of active caries** | No | | | | |
| | Yes | | | 1.09 [0.86–1.39] | 0.463 |
| **Has mobile teeth** | No | | | | |
| | Yes | 1.14 [0.85–1.54] | 0.38 | 1.01 [0.73–1.39] | 0.959 |

*(Continued)*

**Table 3.** (Continued)

| Characteristics | | Hypertension | | Cardiovascular diseases | |
|---|---|---|---|---|---|
| | | OR (95% CI) | P value | OR (95% CI) | P value |
| **Has gum bleeding when brushing teeth** | No | | | | |
| | Yes | 0.91 [0.72–1.15] | 0.433 | **1.69 [1.30–2.21]** | **<0.001** |
| **Has permanent teeth missing due to decay (extracted) and not replaced** | No | | | | |
| | Yes | 1.00 [0.78–1.28] | 0.989 | 0.94 [0.75–1.18] | 0.607 |
| **Oral health** | Optimal | | | | |
| | Suboptimal | | | 1.50 [0.88–2.55] | 0.141 |
| **Presence of prosthetic tooth replacement** | No | | | | |
| | Yes | **1.45 [1.20–1.75]** | **<0.001** | **1.35 [1.07–1.70]** | **0.01** |
| **Last dental Checkup** | More than a year ago | | | | |
| | Less than 6 months ago | 1.03 [0.83–1.27] | 0.798 | 1.11 [0.86–1.44] | 0.415 |
| | less than a year but more than 6 months ago | 0.97 [0.77–1.22] | 0.795 | 0.88 [0.67–1.17] | 0.391 |

Bold values indicate statistical significance at p < 0.05. Adjusted ORs account for other variables within the model. Variables that were excluded by elastic net regularization are shaded with grey.

chronic disease (5.20 [4.36–6.20]) elevated it. Good self-perceived health also reduced the odds while bad health elevated it. Certain oral health conditions increased cardiovascular risk profile odds (mobility: 1.49 [1.10–2.02], prosthetics: 1.69 [1.41–2.03], and recent dental visits 1.30 [1.06–1.60]).

## 3.5 Sensitivity analysis

The results from both bootstrapping and subpopulation analyses consistently affirmed the robustness of the original model. Across various demographic groups and under repeated sampling conditions, the associations originally identified remained stable and unchanged, underscoring the reliability and generalizability of the primary findings.

## 4. Discussion

In light of the study's aim to assess the impact of oral health indicators on hypertension, CVDs, and hypercholesterolemia, findings indicated that certain dental health factors, such as the number of extracted teeth and prosthetic tooth use, were associated with the aforementioned cardiovascular conditions. Poor self-perceived oral health and gum bleeding also emerged as risk factors.

In alignment with findings by Se-Yeon Kim et al. (2018), which advocated for the utility of self-perceived oral health as a reliable indicator of clinical oral health status, particularly in relation to periodontal conditions, the present study provided additional support for the accuracy of self-assessed oral health measures [46]. Furthermore, the work of D. Locker et al. (2000) substantiated the concurrence of poor self-perceived oral health with lower psychological well-being and diminished life satisfaction, particularly among older adults [47]. While these prior studies established the validity of self-perceived oral health as a proxy metric for oral health, they did not explore its relationship with the prevalence of chronic cardiac conditions.

While chi-squared tests indicated significant associations between self-perceived oral health and chronic cardiac conditions, these associations diminished in multiple logistic regression

**Table 4. Weighted multiple logistic regression analysis of factors affecting hypercholesterolaemia and cardiovascular risk profile.**

| Characteristics | | Hypercholesterolaemia | | Cardiovascular risk profile | |
|---|---|---|---|---|---|
| | | OR (95% CI) | P value | OR (95% CI) | P value |
| **Gender** | Male | | | | |
| | Female | 1.13 [0.92–1.40] | 0.242 | 1.10 [0.93–1.30] | 0.257 |
| **Age groups** | 65+ | | | | |
| | 15–34 | **0.27 [0.15–0.50]** | **<0.001** | **0.12 [0.09–0.17]** | **<0.001** |
| | 35–64 | **0.63 [0.48–0.83]** | **0.001** | **0.41 [0.33–0.52]** | **<0.001** |
| **Area of residence** | Rural | | | | |
| | Urban | **1.28 [1.03–1.59]** | **0.025** | 1.13 [0.95–1.35] | 0.153 |
| **Educational attainment** | Primary | | | | |
| | Secondary | 1.10 [0.86–1.39] | 0.458 | 0.87 [0.71–1.07] | 0.183 |
| | Tertiary | **1.40 [1.03–1.89]** | **0.031** | 1.02 [0.79–1.31] | 0.872 |
| **Employment** | Unemployed | | | | |
| | Employed | 1.18 [0.89–1.56] | 0.24 | 0.97 [0.78–1.20] | 0.769 |
| **Financial status** | Average | | | | |
| | Good | 0.96 [0.75–1.23] | 0.75 | | |
| | Bad | 1.09 [0.81–1.46] | 0.559 | | |
| **Income quintiles** | First | | | | |
| | Second | 1.15 [0.86–1.54] | 0.345 | 0.91 [0.71–1.16] | 0.439 |
| | Third | 0.97 [0.70–1.33] | 0.837 | 0.90 [0.69–1.17] | 0.422 |
| | Fourth | 1.22 [0.88–1.68] | 0.226 | 1.09 [0.84–1.42] | 0.508 |
| | Fifth | 1.24 [0.83–1.87] | 0.292 | 0.81 [0.59–1.11] | 0.19 |
| **Smoking** | Smoker | | | | |
| | Non-smoker | 1.04 [0.81–1.34] | 0.742 | | |
| **BMI** | Overweight and Obese | | | | |
| | Normal | **0.51 [0.40–0.65]** | **<0.001** | **0.47 [0.40–0.56]** | **<0.001** |
| **Alcohol use** | Drinker | | | | |
| | Non-drinker | 0.89 [0.71–1.11] | 0.311 | | |
| **Has Chronic disease** | No | | | | |
| | Yes | **3.16 [2.39–4.17]** | **<0.001** | **5.20 [4.36–6.20]** | **<0.001** |
| **Self-perceived health** | Average | | | | |
| | Good | **0.57 [0.44–0.73]** | **<0.001** | **0.53 [0.44–0.64]** | **<0.001** |
| | Bad | **1.67 [1.29–2.16]** | **<0.001** | **1.55 [1.16–2.09]** | **0.003** |
| **Self-perceived oral health** | Average | | | | |
| | Good | 1.11 [0.87–1.40] | 0.403 | 0.95 [0.78–1.15] | 0.595 |
| | Bad | 1.24 [0.96–1.61] | 0.096 | 1.19 [0.94–1.50] | 0.147 |
| **Number of permanent teeth extracted due to decay, not replaced** | None | | | | |
| | 1 to 5 | 1.26 [0.68–2.34] | 0.46 | | |
| | 6 to 19 | 1.26 [0.67–2.35] | 0.473 | | |
| | More than 20 | 1.14 [0.61–2.14] | 0.685 | | |
| **Has filled teeth** | No | | | | |
| | Yes | | | 1.01 [0.83–1.22] | 0.91 |
| **Presence of active caries** | No | | | | |
| | Yes | | | 0.96 [0.78–1.18] | 0.693 |
| **Has mobile teeth** | No | | | | |
| | Yes | 1.36 [1.00–1.87] | 0.053 | **1.49 [1.10–2.02]** | **0.01** |

*(Continued)*

**Table 4.** (Continued)

| Characteristics | | Hypercholesterolaemia | | Cardiovascular risk profile | |
|---|---|---|---|---|---|
| | | OR (95% CI) | P value | OR (95% CI) | P value |
| **Has gum bleeding when brushing teeth** | No | | | | |
| | Yes | **1.40 [1.09–1.81]** | **0.008** | | |
| **Has permanent teeth missing due to decay (extracted) and not replaced** | No | | | | |
| | Yes | 1.00 [0.80–1.24] | 0.964 | 0.90 [0.74–1.09] | 0.264 |
| **Oral health** | Optimal | | | | |
| | Suboptimal | | | 1.21 [0.90–1.61] | 0.209 |
| **Presence of prosthetic tooth replacement** | No | | | | |
| | Yes | **1.51 [1.19–1.92]** | **0.001** | **1.69 [1.41–2.03]** | **<0.001** |
| **Last dental Checkup** | More than a year ago | | | | |
| | Less than 6 months ago | **1.51 [1.19–1.93]** | **0.001** | **1.30 [1.06–1.60]** | **0.013** |
| | less than a year but more than 6 months ago | 1.08 [0.82–1.43] | 0.569 | 0.87 [0.70–1.09] | 0.226 |

Bold values indicate statistical significance at p < 0.05. Adjusted ORs account for other variables within the model. Variables that were excluded by elastic net regularization are shaded with grey.

models after adjusting for confounders. This suggests that other variables may influence the relationship between oral health and chronic cardiac conditions, highlighting the complexity of oral-systemic health connections.

This study augments existing literature by showing a notable association between self-perceived oral health and chronic cardiac conditions. Generally, individuals without these conditions reported better oral health, while those with them often reported worse, supporting the concept of a bidirectional relationship between oral and systemic health.

The observed elevated odds for hypertension among individuals with 6 to 19 teeth extracted might intersect with the phenomenon related to prosthetic dental replacements increasing hypertension odds as well, in several ways.

Firstly, both conditions, multiple teeth extractions and the need for prosthetic replacements, could signify a history of chronic oral infections, tooth decay and periodontal disease. These conditions can induce a state of systemic inflammation, which is in line with what Ramadan et al. (2020), and Passarelli et al. (2020), have concluded on the given topic [48, 49]. The bioactive molecules released during this chronic low-grade inflammation, such as cytokines, can adversely affect vascular endothelial function, setting the stage for increased arterial stiffness and hypertension [50].

Secondly, multiple teeth extractions often necessitate prosthetic replacements for functional reasons, such as chewing and speech. Research by Kim et al. (2021) found that the presence of 6–10 and 11 prosthetic crowns increased the prevalence of periodontitis by 1.24 and 1.28 times, respectively, compared to individuals without any prosthetic crowns [51]. This suggests that prosthetic replacements may serve as foci for continued bacterial infection and periodontal disease. Consequently, these localized oral health issues could perpetuate the cycle of systemic inflammation, thereby elevating hypertensive risk [52, 53].

Furthermore, severe tooth loss can impact nutrition. Kossioni (2018) highlighted that tooth loss often leads to dietary changes, with increased consumption of processed foods high in

sodium and unhealthy fats, and reduced intake of fibrous fruits and vegetables, contributing to hypertension risk [54, 55].

Lastly, these conditions may reflect healthcare-seeking behaviour. Older individuals with multiple extractions and prosthetics are likely under regular healthcare supervision and may use medications that influence blood pressure.

Oral health indicators, notably gum bleeding and the presence of prosthetic dental replacements, were significantly associated with increased CVD risk in this study. Extensive tooth loss, particularly more than 20 missing teeth, was also common among individuals with CVD, supporting the notion of systemic inflammation and endothelial dysfunction as overlapping mechanisms linking oral and cardiovascular health. Dhotre et al. (2016) highlighted how periodontitis could contribute to atherosclerotic plaque formation by allowing oral bacteria to enter the bloodstream [56]. This aligns with Ola Vedin et al. (2014), who found a correlation between periodontal disease indicators, such as gum bleeding, and elevated cardiovascular risk [57]. Prosthetic dental replacements may serve as persistent foci for bacterial accumulation, thereby contributing to atherosclerotic plaque formation. The widespread tooth loss noted in the CVD cohort may reflect a nutritional landscape marked by lipid abnormalities and diminished cardiovascular health. Collectively, these findings substantiate a complex, bidirectional interaction between oral health parameters and CVD risk, thereby augmenting current scientific discourse on the systemic implications of oral health.

The significance of these findings is particularly relevant in Hungary, where chronic conditions and heart disease-related mortality are prevalent. Given Hungary's high rates of gingivitis and periodontal disease, these results contribute to both the global understanding of CVD and offer critical insights for its management and prevention within the Hungarian context.

In accordance with the existing scientific literature, the present study delineated a statistically significant elevated risk of hypercholesterolaemia among urban residents and individuals with higher educational attainment [58–64].

The study also revealed specific oral health indicators notably influencing the risk profile for hypercholesterolaemia. Gingival bleeding was associated with elevated odds of hypercholesterolemia, as was the presence of prosthetic dental replacements. This association is supported by extant literature explaining the biological linkage between periodontal symptoms and elevated plasma cholesterol levels [65, 66]. For instance, Tahamtan et al. (2020) documented the salutary effects of statins on various dimensions of oral health, including chronic periodontitis [67]. Additionally, a 2014 study indicated that periodontal therapy ameliorated subclinical arterial thickness, implicating a correlation between atheromatous plaques and periodontal disease [68]. Fu et al. (2016) further substantiated this link, demonstrating that periodontal treatment led to improved serum lipid and proinflammatory cytokine profiles in hyperlipidaemic patients [69].

In a departure from established literature, the current study noted two interesting observations, one is the association between recent dental visits (within the last 6 months) and an elevated risk of hypercholesterolemia. Interpretation could involve the concept of "risk factor clustering". Individuals who seek dental care frequently may already be engaged in healthcare surveillance due to other comorbidities or risk factors, including dyslipidaemia. Their dental symptoms might be reflective of underlying systemic inflammation, which is also a key player in the pathophysiology of hypercholesterolemia. The elevated lipid levels could be both a cause and a consequence of inflammatory oral conditions, such as periodontitis, thus leading to a cyclical relationship between oral health and lipid metabolism. The other observation was the lack of statistically significant link between dental caries and CVDs, which was discussed in a previous review article [70]. This could potentially be attributed to factors such as the heterogeneity in dental caries progression and treatment, and the multifactorial nature of CVD,

which involves various risk factors that could obscure the impact of caries alone. Additionally, the criteria for defining dental caries might not fully reflect the disease's severity or its microbial aspects, like S. mutans, suggesting the need for more focused research on the specific interactions between oral health and cardiovascular diseases.

### 4.1 Strengths and limitations

This study distinguishes itself not only through its robust methodological framework but also by filling a conspicuous gap in the literature on the Central and Eastern European region. To the authors' knowledge, it is the inaugural study of its kind in the Hungarian population to study the relationship between numerous chronic non-communicable diseases and oral health. The utilization of data from the EHIS lends considerable weight to the generalizability of the study's findings. Methodological rigor is further accentuated by the incorporation of multiple socioeconomic and demographic confounders, comprehensive data protocols, and innovative machine learning applications for variable selection and sensitivity analyses.

This study has limitations, primarily due to its cross-sectional design which limits causal inferences and temporal relationship establishment. Reliance on self-reported data raises concerns of recall bias, affecting metrics like comorbidities and BMI accuracy. Furthermore, the study's scope was constrained by not differentiating chronic disease severity and excluding key biochemical parameters (e.g., glycemia, HbA1c), limiting insights into metabolic and renal impacts on cardiovascular outcomes. Also, while most variables exhibited high levels of data completeness, a few had a noticeable proportion of missing values. This could potentially influence the accuracy and generalizability of the findings related to these specific aspects.

## 5. Conclusion

In a nationally representative sample of the Hungarian population, suboptimal oral health markers were found to be significantly associated with increased risk for negative cardiac outcomes. Conversely, improvements in oral health parameters are linked to a diminution in the risk profiles of these outcomes. Targeted oral health interventions could serve as a vital component of a multi-disciplinary approach to disease management and prevention. Future research should delve into the causal and temporal relationships between oral health and heart and vascular conditions, incorporating clinical assessments alongside self-reported data to enhance accuracy and consider medication use as a potential confounder. Additionally, examining the efficacy of oral health interventions on disease outcomes and dissecting variability across demographic groups could yield critical insights for tailored public health strategies.

## Supporting information

**S1 Table. Bootstrap analysis of oral health indicators and their association with cardiovascular outcomes.**
(DOCX)

**S2 Table. Elastic net selection results with α = 0.5: Predictors for hypertension and cardiovascular diseases.**
(DOCX)

**S3 Table. Elastic net selection results with α = 0.5: Predictors for hypercholesterolaemia and cardiovascular risk profile.**
(DOCX)

**S1 Fig. Flowchart of the study participants and variable groupings.**
(TIF)

## Author Contributions

**Conceptualization:** Amr Sayed Ghanem, Attila Csaba Nagy.

**Formal analysis:** Amr Sayed Ghanem.

**Methodology:** Amr Sayed Ghanem, Attila Csaba Nagy.

**Supervision:** Attila Csaba Nagy.

**Writing – original draft:** Amr Sayed Ghanem.

**Writing – review & editing:** Amr Sayed Ghanem, Orsolya Németh, Marianna Móré, Attila Csaba Nagy.

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
