## [Decision Letter · Decision Letter 0]

13 Feb 2024

PONE-D-23-40949Role of oral health in heart and vascular health: A population-based studyPLOS ONE

Dear Dr. Nagy,

Thank you for submitting your manuscript to PLOS ONE. After careful consideration, we feel that it has merit but does not fully meet PLOS ONE’s publication criteria as it currently stands. Therefore, we invite you to submit a revised version of the manuscript that addresses the points raised during the review process.

We look forward to receiving your revised manuscript.

Kind regards,

Artak Heboyan, Ph.D.

Academic Editor

PLOS ONE

Journal Requirements:

3. Please upload a copy of Figure 1, to which you refer in your text on page 5. If the figure is no longer to be included as part of the submission please remove all reference to it within the text.

Reviewers' comments:

Reviewer's Responses to Questions

**Comments to the Author**

1. Is the manuscript technically sound, and do the data support the conclusions?

Reviewer #1: Yes

Reviewer #2: Yes

Reviewer #3: Yes

Reviewer #4: Yes

Reviewer #5: Yes

2. Has the statistical analysis been performed appropriately and rigorously? 

Reviewer #1: I Don't Know

Reviewer #2: Yes

Reviewer #3: Yes

Reviewer #4: Yes

Reviewer #5: Yes

3. Have the authors made all data underlying the findings in their manuscript fully available?

Reviewer #1: Yes

Reviewer #2: Yes

Reviewer #3: Yes

Reviewer #4: Yes

Reviewer #5: Yes

4. Is the manuscript presented in an intelligible fashion and written in standard English?

Reviewer #1: Yes

Reviewer #2: Yes

Reviewer #3: Yes

Reviewer #4: Yes

Reviewer #5: Yes

5. Review Comments to the Author

Reviewer #1: A review report of the manuscript titled "Role of oral health in heart and vascular health: A population-based study".I read this paper with interest and found that mostly it is well performed and well-written. However, the discussion part still needs some further improvement. Authors should perform more extensive comparisons. I recommend to add some references: https://opendentistryjournal.com/VOLUME/16/ELOCATOR/e187421062209270/

Good luck

Reviewer #2: A diligently prepared study. I would like to express my gratitude to the authors for their efforts. Below are a few suggestions regarding the study.

In introduction “The etiological landscape is further complicated by a many of factors…” -- > “…by many factors…”

The keyword "dental caries" is provided, but in the study, the relationship between dental caries and cardiovascular disease (CVD) was found to be insignificant. While a significant relationship between S. mutans found in caries microbiology and CVD has been established in the literature, this discrepancy could be briefly discussed. (Nakano, Kazuhiko, Ryota Nomura, and Takashi Ooshima. "Streptococcus mutans and cardiovascular diseases." Japanese Dental Science Review 44.1 (2008): 29-37).

Reviewer #3: This study titled ‘Role of oral health in heart and vascular health: A population-based study” aimed to assess the impact of oral health indicators on hypertension, CVDs, and hypercholesterolemia among the Hungarian population.

The findings of the manuscript revealed that suboptimal oral health markers were found to be significantly associated with increased risk for negative cardiac outcomes. Conversely, improvements in oral health parameters are linked to a diminution in the risk profiles of these outcomes.

Overall, the manuscript is clear and very well written and is of importance considering that it fills a gap in the literature on the Central and Eastern European region.

Reviewer #4: 1. in the material and methods section:

Please add information regarding inclusion and exclusion criteria for selecting data included in the analysis. For example, the range of age included.

2. please add information about how the author measures hypertension, hypercholesterolemia,CV disease from the survey

3 How do you measure oral health in the questionnaire?

4.How did the author measure variable of chronic disease?

5.How did the author measure the variables of activities of caries

please include more information in the manuscript regarding the measurement of those variables

6.Based on your study, what can you suggest for further study to elaborate the findings of your study please information regarding this

Reviewer #5: This is an interesting article that is generally well written.

A minor comment on ensuring that causality is not inferred in the way you describe your conclusions.

E.g. in the Discussion section, it is important to ensure that causality is not attributed to the interpretation of the results. '

E.g. “CVDs, and hypercholesterolemia, findings indicated that certain dental health factors, such as the number of extracted teeth and prosthetic tooth use, were associated with increased risks of these conditions.”

could be written as:

“certain dental health factors, such as the number of extracted teeth and prosthetic tooth use, were associated with CVD diagnosis and hypercholesterolemia”

Well done!

6. PLOS authors have the option to publish the peer review history of their article (what does this mean?). If published, this will include your full peer review and any attached files.

Reviewer #1: No

Reviewer #2: **Yes: **Fatih ŞENGÜL

Reviewer #3: No

Reviewer #4: **Yes: **kusdhany Lindawati S

Reviewer #5: No

---

## [Author Response · Author response to Decision Letter 0]

28 Feb 2024

Dear Editor,

Thank you for the opportunity to submit a revised draft of our manuscript. We appreciate the time and effort you and the reviewers have taken to provide feedback on our manuscript. We are grateful to the reviewers for their valuable insights, and we have incorporated their suggestions into the revised version of the manuscript.

Please find below our response to both your comments, and the reviewers' comments and concerns.

Editor comment: Your ethics statement should only appear in the Methods section of your manuscript. If your ethics statement is written in any section besides the Methods, please move it to the Methods section and delete it from any other section. Please ensure that your ethics statement is included in your manuscript, as the ethics statement entered into the online submission form will not be published alongside your manuscript.

Response: We have deleted our ethics statement from the very end of the manuscript, and placed it in the Methods section as requested.

Editor comment: Please upload a copy of Figure 1, to which you refer in your text on page 5. If the figure is no longer to be included as part of the submission please remove all reference to it within the text.

Response: A pdf filed titled S1_fig has been uploaded alongside the resubmitted manuscript, containing the flowchart referenced in the manuscript.

Editor comment: Please include captions for your Supporting Information files at the end of your manuscript, and update any in-text citations to match accordingly.

Response: Captions for the supporting information files, including the file containing the three supporting tables, and the pdf file containing the figure, has been added to the appropriate section in the manuscript as per the journal guidelines.

Comments from Reviewer 1

Dear Reviewer,

Thank you for your time and consideration in reviewing our submission. We appreciate your valuable feedback and suggestions for improving the quality of our manuscript. We remain available to answer any further questions or concerns you may have.

Reviewer comment: A review report of the manuscript titled "Role of oral health in heart and vascular health: A population-based study".I read this paper with interest and found that mostly it is well performed and well-written. However, the discussion part still needs some further improvement. Authors should perform more extensive comparisons. I recommend to add some references: https://opendentistryjournal.com/VOLUME/16/ELOCATOR/e187421062209270/

Good luck

Response: Thank you very much for your insightful comment and the suggestion to include additional references to strengthen our manuscript. We appreciate the opportunity to review the literature you recommended and found it to be a valuable addition to our discussion. Accordingly, we have cited this work in the revised manuscript, citation number 64, to further provide a more comprehensive perspective on the subject matter. We believe this addition enhances the depth of our discussion and are grateful for your contribution to improving our work.

Comments from Reviewer 2

Dear Reviewer,

Reviewer comment: A diligently prepared study. I would like to express my gratitude to the authors for their efforts. Below are a few suggestions regarding the study.

In introduction “The etiological landscape is further complicated by a many of factors…” -- > “…by many factors…”

The keyword "dental caries" is provided, but in the study, the relationship between dental caries and cardiovascular disease (CVD) was found to be insignificant. While a significant relationship between S. mutans found in caries microbiology and CVD has been established in the literature, this discrepancy could be briefly discussed. (Nakano, Kazuhiko, Ryota Nomura, and Takashi Ooshima. "Streptococcus mutans and cardiovascular diseases." Japanese Dental Science Review 44.1 (2008): 29-37).

Response: Thank you for your kind words and constructive suggestions. We have corrected the typographical error in the introduction as you recommended. Additionally, we have addressed the discrepancy regarding the relationship between dental caries, and cardiovascular disease highlighted in the discussion section of our revised manuscript. We appreciate your guidance in enhancing the clarity and depth of our study.

Comments from Reviewer 3

Dear Reviewer,

Reviewer #3: This study titled ‘Role of oral health in heart and vascular health: A population-based study” aimed to assess the impact of oral health indicators on hypertension, CVDs, and hypercholesterolemia among the Hungarian population.

The findings of the manuscript revealed that suboptimal oral health markers were found to be significantly associated with increased risk for negative cardiac outcomes. Conversely, improvements in oral health parameters are linked to a diminution in the risk profiles of these outcomes.

Overall, the manuscript is clear and very well written and is of importance considering that it fills a gap in the literature on the Central and Eastern European region.

Response: We sincerely appreciate your thoughtful and encouraging comments regarding our manuscript. We are gratified to learn that the manuscript's findings and its contribution to understanding the impact of oral health on cardiovascular diseases within the Hungarian population resonate with its intended purpose. Your positive feedback serves as a great motivation for our team and reinforces the importance of our research.

Comments from Reviewer 4

Dear Reviewer,

Reviewer comments: 1. in the material and methods section:

Please add information regarding inclusion and exclusion criteria for selecting data included in the analysis. For example, the range of age included.

2. please add information about how the author measures hypertension, hypercholesterolemia,CV disease from the survey

3 How do you measure oral health in the questionnaire?

4.How did the author measure variable of chronic disease?

5.How did the author measure the variables of activities of caries

please include more information in the manuscript regarding the measurement of those variables

6.Based on your study, what can you suggest for further study to elaborate the findings of your study please information regarding this

Response: Thank you for your valuable comments. In our analysis, all participants aged 15 and above were included, with age serving as a key variable, stratified into distinct groups as outlined in the methods and the results. This stratification allowed us to treat each age group as a separate stratum within the logistic regression models, enabling us to conduct detailed analyses of how chronic disease outcomes varied across different age categories. By analyzing these age strata individually, supported by the subpopulation analysis, we were able to capture differences and assess the impact of age on chronic disease outcomes across different age groups with greater specificity. 

In response to your feedback, we've updated the methods section to clarify our variable selection criteria, emphasizing their theoretical and empirical basis. We hope this adjustment addresses your concerns and refines the manuscript's methodological detail. You can find the updated section highlighted in page 6 of our revised manuscript.

Regarding your inquiries for the measurement of hypertension, hypercholesterolemia, cardiovascular disease, oral health indicators, variables of chronic disease, and presence of active caries, we would like to clarify that these variables were directly assessed through the European Health Interview Survey questionnaire. Participants were asked to report whether they suffer from these conditions, providing a self-reported measure of each. This method of data collection is consistent with the survey's methodology, designed to capture a wide range of health indicators through participant self-reporting. For further detail, the survey questionnaire, distributed by the Central Statistical Office of Hungary, is cited in our manuscript (citation number 40). To facilitate a more comprehensive review, we are also including a link to the questionnaire for your convenience.

https://www.ksh.hu/elef/elef2019_kerdoiv.pdf , We hope this addresses your queries satisfactorily.

In response to the findings of our study, we suggest several avenues for future research to further elaborate on our results. Firstly, exploring the causal and temporal mechanisms behind the associations we observed could provide deeper insights into the relationship between oral health and cardiac outcomes. Additionally, while our study utilized self-reported data from the European Health Interview Survey, future research could benefit from incorporating clinical assessments to validate these self-reports and provide a more detailed understanding of health status, also factoring in medication use as a confounder.

Moreover, investigating the impact of specific oral health interventions on chronic disease outcomes could offer valuable information for public health strategies. Our findings also indicate the potential for significant variability in outcomes across different demographic groups; thus, future studies could focus on understanding these disparities in greater detail.

Based on your suggestion, we expanded the conclusion section of our manuscript to appropriately reflect directions for future research. We believe that your suggestions greatly contributed to the rigor and quality of our manuscript, thank you again.

Comments from Reviewer 5

Dear Reviewer,

Reviewer comment: A minor comment on ensuring that causality is not inferred in the way you describe your conclusions.

E.g. in the Discussion section, it is important to ensure that causality is not attributed to the interpretation of the results. '

E.g. “CVDs, and hypercholesterolemia, findings indicated that certain dental health factors, such as the number of extracted teeth and prosthetic tooth use, were associated with increased risks of these conditions.”

could be written as:

“certain dental health factors, such as the number of extracted teeth and prosthetic tooth use, were associated with CVD diagnosis and hypercholesterolemia”

Well done!

Response: Thank you for your valuable feedback. The phrasing has been updated as per your suggestion; you can find the modification highlighted in page 15 of the revised manuscript. Thank you for your time and effort in reviewing our work.

Once again, we extend our sincere thanks for your thorough review and insightful comments, which have been vital in increasing the quality and clarity of our manuscript. We are hopeful that the revisions and explanations provided address your concerns and contribute to the strength of our study. 

Best regards,

Dr. Attila Csaba Nagy

On behalf of the authors

---

## [Decision Letter · Decision Letter 1]

18 Mar 2024

Role of oral health in heart and vascular health: A population-based study

PONE-D-23-40949R1

Dear Dr. Nagy,

We’re pleased to inform you that your manuscript has been judged scientifically suitable for publication and will be formally accepted for publication once it meets all outstanding technical requirements.

Kind regards,

Artak Heboyan, Ph.D.

Academic Editor

PLOS ONE

Additional Editor Comments (optional):

Reviewers' comments:

Reviewer's Responses to Questions

**Comments to the Author**

1. If the authors have adequately addressed your comments raised in a previous round of review and you feel that this manuscript is now acceptable for publication, you may indicate that here to bypass the “Comments to the Author” section, enter your conflict of interest statement in the “Confidential to Editor” section, and submit your "Accept" recommendation.

Reviewer #4: All comments have been addressed

Reviewer #5: (No Response)

2. Is the manuscript technically sound, and do the data support the conclusions?

Reviewer #4: Yes

Reviewer #5: Yes

3. Has the statistical analysis been performed appropriately and rigorously? 

Reviewer #4: Yes

Reviewer #5: Yes

4. Have the authors made all data underlying the findings in their manuscript fully available?

Reviewer #4: Yes

Reviewer #5: Yes

5. Is the manuscript presented in an intelligible fashion and written in standard English?

Reviewer #4: Yes

Reviewer #5: Yes

6. Review Comments to the Author

Reviewer #4: (No Response)

Reviewer #5: By passing this section as all comments originally provided to authors have been addressed.

I have not competing interests, and no further comments.

7. PLOS authors have the option to publish the peer review history of their article (what does this mean?). If published, this will include your full peer review and any attached files.

Reviewer #4: No

Reviewer #5: No

---

## [Editor Report · Acceptance letter]

21 Mar 2024

PONE-D-23-40949R1 

PLOS ONE

Dear Dr. Nagy, 

I'm pleased to inform you that your manuscript has been deemed suitable for publication in PLOS ONE. Congratulations! Your manuscript is now being handed over to our production team.

Kind regards, 

on behalf of

Dr. Artak Heboyan 

Academic Editor

PLOS ONE